# Dynamics of Actin Cytoskeleton and Their Signaling Pathways during Cellular Wound Repair

**DOI:** 10.3390/cells11193166

**Published:** 2022-10-09

**Authors:** Shigehiko Yumura, Md. Shahabe Uddin Talukder, Mst. Shaela Pervin, Md. Istiaq Obaidi Tanvir, Takashi Matsumura, Koushiro Fujimoto, Masahito Tanaka, Go Itoh

**Affiliations:** 1Graduate School of Sciences and Technology for Innovation, Yamaguchi University, Yamaguchi 753-8511, Japan; 2Institute of Food and Radiation Biology, AERE, Bangladesh Atomic Energy Commission, Savar, Dhaka 3787, Bangladesh; 3Rajshahi Diabetic Association General Hospital, Luxmipur, Jhautala, Rajshahi 6000, Bangladesh; 4Department of Molecular Medicine and Biochemistry, Akita University Graduate School of Medicine, Akita 010-8543, Japan

**Keywords:** actin, actin-binding proteins, cell membrane, signaling, wound repair

## Abstract

The repair of wounded cell membranes is essential for cell survival. Upon wounding, actin transiently accumulates at the wound site. The loss of actin accumulation leads to cell death. The mechanism by which actin accumulates at the wound site, the types of actin-related proteins participating in the actin remodeling, and their signaling pathways are unclear. We firstly examined how actin accumulates at a wound site in *Dictyostelium* cells. Actin assembled de novo at the wound site, independent of cortical flow. Next, we searched for actin- and signal-related proteins targeting the wound site. Fourteen of the examined proteins transiently accumulated at different times. Thirdly, we performed functional analyses using gene knockout mutants or specific inhibitors. Rac, WASP, formin, the Arp2/3 complex, profilin, and coronin contribute to the actin dynamics. Finally, we found that multiple signaling pathways related to TORC2, the Elmo/Doc complex, PIP2-derived products, PLA2, and calmodulin are involved in the actin dynamics for wound repair.

## 1. Introduction

The cell membrane is a barrier between the extracellular and intracellular spaces, but it is consistently subjected to wounding by physical or chemical damage. However, wounded cell membranes can be repaired. Loss of wound repair is seen in various diseases such as muscular dystrophy [1,2]. The molecular mechanisms of wound repair have been studied in different model organisms, such as mammalian cells [3], amphibian eggs [4,5], echinoderm eggs [6], fruit flies [7,8,9], nematodes [1,10], amoebae [11], yeast [12], and *Dictyostelium* cells [13]. A common feature among these mechanisms is that Ca^2+^ influx from an external medium is essential for wound repair. The “membrane patch hypothesis” has been proposed to explain wound pore plugging, wherein cytosolic membrane vesicles accumulate at the wound site and fuse to form an impermanent “patch” to plug the wound pore as an emergency reaction [6,14,15]. The source of the membrane plug is controversial, and we recently proposed that the vesicles for the patch are synthesized de novo [16]. Various hypotheses for wound repair that do not involve patching have also been proposed [3,17,18]. However, there is no clear consensus on the mechanisms that drive the repair process.

In large cells, such as *Xenopus* eggs and *Drosophila* embryos, an actomyosin ring, similar to the contractile ring in dividing cells, surrounds the wound site in the cell membrane, and its constriction facilitates the closure of the wound pore [7,19,20]. However, in smaller cells, such as yeast cells, animal culture cells, and *Dictyostelium* cells, only actin transiently accumulates at the wound site [12,21,22,23]. Under the loss of actin polymerization, wound pores do not close in *Dictyostelium* cells, mammalian culture cells, and muscle cells [16,24,25]. In contrast, myosin II does not accumulate at the wound site, and its contribution to wound repair in small cells is controversial [12,22,23,26]. Myosin II transiently disappears from the wound site in *Dictyostelium* cells, and the deletion of myosin II does not affect wound repair [27]. Similar to other models for wound repair, annexins, which are membrane scaffold proteins and/or endosomal sorting complexes required for transport (ESCRT), are also candidates for the wound repair mechanism; these proteins may contribute to the closure of wound pores or the cutting off the wounded membrane [28,29].

The mechanism by which actin accumulates to form specific actin structures in specific cortical regions, such as pseudopodia and filopodia, has been intensively investigated. Active Rac and Cdc42, small Rho-family GTPases, activate Wiskott–Aldrich Syndrome protein (WASP) family proteins, such as WASP, SCAR, and WASH. In turn, these WASP family proteins activate the Arp2/3 complex, which generates a meshwork of branched F-actin [30]. Cdc42 and RhoA activate formin, which nucleates linear actin filaments [31].

In addition to Arp2/3 and formin, many species of actin-related proteins (ARPs) regulate actin organization and its dynamics, and are categorized as crosslinking, monomer-binding, bundling, stabilizing, branch-forming, and membrane-binding [32]. To form a specific actin structure in specific regions, specific members of ARPs should be spatially and temporally recruited and regulated. However, such information is limited, particularly for wound repair.

As mediators of signaling pathways for wound-induced actin assembly, WASP, Arp2/3, and formin have been implicated in regulating actin assembly [33,34,35,36]. We recently found that Ca^2+^ and calmodulin regulate actin assembly in *Dictyostelium* cells [27]. Protein kinase C (PKC) is also involved in actin assembly in *Xenopus* oocytes and yeast during wound repair [12,37]. Rho families and their upstream regulators also regulate the wound-induced actomyosin ring in *Drosophila* embryos and *Xenopus* oocytes [8,38]. In addition, annexin A2 and S100A11 complexes regulate actin assembly at wound sites in human cells [24,39].

The signaling pathways of actin assembly in the chemotaxis of *Dictyostelium* cells as well as neutrophils have been intensively investigated [40,41]. The binding of cAMP, a chemoattractant for *Dictyostelium*, to its membrane receptors, which are trimeric G protein-coupled receptors (GPCRs), causes actin assembly at the leading edge to extend pseudopods. Several pathways are involved in actin assembly and chemotaxis, including phosphatidylinositol 3,4,5-trisphosphate kinase (PI3K)-, phospholipase A2 (PLA2)-, cGMP-, TORC2-, and Elmo/Doc complex-mediated pathways, which seem to form a cross-talk network [42,43,44]. Actin assembly for chemotaxis and wound repair might share common signals.

In this study, we examined how actin accumulates at a wound site and the types of ARPs that participate in actin remodeling. With reference to the signal cascades for chemotaxis, we also examined the types of signals that regulate the dynamics of these proteins and found several candidates for the signaling pathways. This information should also provide insights into the mechanism not only for actin organization during wound repair but also for those during many actin-dependent cellular events such as chemotaxis, phagocytosis, macropinocytosis, and cell adhesions.

## 2. Materials and Methods

### 2.1. Cell Culture

*Dictyostelium discoideum* wild-type (AX2) and mutant cells were cultured at 22 °C in a plastic dish containing HL5 medium (1.3% bacteriological peptone, 0.75% yeast extract, 85.5 mM d-glucose, 3.5 mM Na_2_HPO_4_, and 3.5 mM KH_2_PO_4_, pH 6.3), as previously described [45]. For wound experiments, the HL5 medium was replaced with BSS (10 mM NaCl, 10 mM KCl, 3 mM CaCl_2_, and 3 mM MES, pH 6.3), and the cells were incubated in the same solution for 5–6 h.

### 2.2. Plasmids and Mutants

Descriptions of the plasmid sources and null mutant cells used in the present study are listed in Appendix A. Cells were transformed with expression vectors of GFP-tagged proteins were transformed into cells by electroporation or laserporation, as described previously [46,47]. The transformed cells were selected in HL5 medium containing 10 µg/mL G418 (Wako Pure Chemical Corporation, Osaka, Japan) or 10 µg/mL blasticidin S hydrochloride (Wako Pure Chemical Corporation, Osaka, Japan) in plastic dishes.

We constructed expression vectors for some GFP-tagged proteins, including RacA, severin, filamin, ABD34, CAP32, fimbrin, α-actinin, and CARMIL. The individual genes were amplified from a cDNA library by PCR and subcloned between the BamH1 and Sac1 sites downstream of the C-terminal GFP site in the pA15GFP expression vector, as described previously [16].

### 2.3. Chamber Preparation

The surface of the coverslip of a glass-bottom chamber was coated with gold using a sputtering device (DC-704 Quick Coater, Sanyu Electron Co. Ltd., Tokyo, Japan), in place of carbon vapor deposition, as previously described [47,48]. The thickness of the coating layer was approximately 20 nm. To render the surface hydrophilic, the surface of the coated coverslip was activated by plasma treatment. The chamber was sterilized with 70% ethanol and dried if necessary. The cells were placed on the surface of the coated coverslip and mildly compressed with an agarose block (2%, dissolved in BSS, 1 mm thick) to observe the ventral cell surface [49,50].

### 2.4. Wounding and Microscopy

Cells expressing GFP-tagged proteins were observed under a total internal reflection fluorescence (TIRF) microscope (based on the IX71 microscope, Olympus, Tokyo, Japan), as previously described [51]. The cells were wounded with a nanosecond-pulsed laser (FDSS532-Q, CryLas, Berlin, Germany), and the wound diameter was set as 0.5 µm, as previously described [13]. Time-lapse fluorescence images were acquired at 40–100 ms exposure and 130–500 ms intervals using a cooled CCD camera (Orca ER, Hamamatsu Photonics, Shizuoka, Japan). The time courses of the fluorescence intensities within the circle (3 µm in diameter), including the wound site, were examined using ImageJ software (http://rsbweb.nih.gov/ij, accessed on 1 September 2022). The fluorescence intensities were normalized by setting the value before wounding to 1 after subtracting the background.

### 2.5. Measurement of PI Influx

To observe the influx of PI, a 0.15 mg/mL propidium iodide (PI) solution (Sigma–Aldrich, Tokyo, Japan) was added to the external medium before wound experiments. PI fluorescence was monitored by illumination with an argon laser (488 nm) under TIRF microscopy, as described before [13].

### 2.6. Inhibitors

Latrunculin A, jasplakinolide, W7 hydrochloride, CK666, CK869, SMIFH2, EHT1864, wiscostatin, Torin 1, U73122, and LY294002 were purchased from Funakoshi Co., Ltd. (Tokyo, Japan). BPB was purchased from the FUJIFILM Wako Pure Chemical Corporation (Osaka, Japan). Thiabendazole was purchased from Tokyo Chemical Industry Co., Ltd. (Tokyo, Japan). All inhibitors were dissolved in dimethyl sulfoxide (DMSO) to prepare stock solutions and kept at −30 °C until use. Cells were incubated with the following concentrations of inhibitors: 1 µM latrunculin A, 8 µM jasplakinolide, 20 µM W7 hydrochloride, 60 µM CK666, 50 µM CK869, 10 µM SMIFH2, 50 µM EHT1864, 10 µM wiscostatin, 50 µM Torin 1, 4 µM U7312220, 20 µM LY294002, 2 µM BPB, and 100 µM thiabendazole. Inhibitors were used at a higher than IC50 concentration. The IC50 concentrations of CK666, U731220, BPB, wiscostatin, and torin1 were based on macropinocytosis inhibition in previous report [52]. The concentrations of latrunculin A, jasplakinolide, W7 hydrochrolide, and thiabendazole were used as previously described [16]. The IC50 concentrations of CK869, SMIFH2, EHT1864 were 30, 20, 20 µM, respectively, which were determined by examining cell velocities. The final concentration of DMSO did not exceed 0.5% (*v*/*v*), which did not affect wound repair, as described before [23]. Wound experiments were conducted 30 min after incubation with these inhibitors.

### 2.7. Statistical Analysis

Statistical analyses were performed using GraphPad Prism 7 (GraphPad Software Inc., San Diego, CA, USA). Data were expressed as means ± SD and analyzed using a two-tailed Student’s *t*-test for comparison between two groups, or one-way ANOVA with Tukey’s multiple comparisons test.

## 3. Results

### 3.1. Improved Laserporation Using Gold Coating

We previously reported a novel method for creating wounds in cell membranes using laserporation [13,46]. In this study, the coverslip was coated with gold instead of carbon. After the cells were placed on the coverslip coated with gold via vapor deposition, a laser beam was focused on a small local spot beneath a single cell under a TIRF microscope. The energy of the laser beam absorbed by the gold made a small pore in the cell membrane that was attached to the gold coating (Figure 1A). The gold coating absorbed less fluorescence than the carbon coating; therefore, fluorescence imaging was greatly improved.

It was easy to focus the laser spot on the surface of the coverslip. When the laser beam was thus focused, the gold coating peeled off and appeared as a small white spot (Figure 1B). Because the laser power used for the actual wound experiments was much lower, the gold coating did not peel off. Therefore, it was also possible to create multiple wounds at the same site.

### 3.2. Actin Accumulation at Wound Site Is Essential for Wound Repair

We used the improved laserporation method to examine the wound repair mechanism in *Dictyostelium*, a model organism for studying cell migration, chemotaxis, and cell division. To examine the opening of a pore in the cell membrane by laserporation, PI, which emits fluorescence upon binding to RNA or DNA, was added to the external medium (BSS). Figure 1C shows the typical time course of the fluorescence images of the PI influx. Fluorescence began to increase at the wound site and spread over the cytoplasm, suggesting that PI enters the cell through the wound pores. Figure 1D (BSS) shows the time course of the fluorescence intensity of PI in the wounded cells, suggesting that PI influx ceased within 2–3 s after wounding, and that urgent wound repair terminated within this time.

We have shown that actin transiently accumulates at the wound site in *Dictyostelium* cells [16,23,47]. Figure 1E shows a typical time course of fluorescence images under TIRF microscopy when a cell expressing GFP-lifeact, a marker of actin filaments, was wounded. Actin filaments transiently accumulated at the wound site (arrow), confirming previous observations using carbon-coated substrates [16]. Previous studies on cultured mammalian cells have indicated that cortical actin networks around wound pores are largely removed before actin accumulation [36,53,54,55]. Under TIRF microscopy, cortical actin was observed as a meshwork composed of filaments [51,56]. Figure 1F shows enlarged images of the rectangle in panel E before wounding (−0.5 s) and 0.5 and 3 s after wounding, and the merged images. The removal of pre-existing cortical actin networks was substantially undetectable, contrary to observations in mammalian cells.

Figure 1G (BSS) shows the time course of the relative fluorescence intensity of GFP-lifeact at the wound site, which was similar to that of GFP-actin, as previously shown [16]. In the presence of latrunculin A, a depolymerizer of actin filaments, actin did not accumulate upon wounding (LatA in Figure 1G) and PI influx did not cease, in contrast to that in the control (LatA in Figure 1D). In addition, we previously reported that the wound hole did not close in the presence of latrunculin A by staining the cell membrane [16].

Therefore, actin accumulation at the wound site is essential for wound repair.

### 3.3. Actin Polymerizes De Novo at Wound Site

There are two possible mechanisms for actin accumulation at a wound site: (1) preexisting cortical actin filaments flow (moving along the cell membrane) to accumulate toward the wound site (flow model), and (2) actin polymerizes at the wound site (de novo synthesis model). In *Xenopus* oocytes, the actomyosin ring is generated by the flow of cortical actin filaments toward the wound site, accompanied by dynamic actin polymerization [19,57]. However, we did not observe such a flow around the wound site (Figure 1F), despite the ability of TIRF microscopy to enable the direct observation of individual actin filaments.

To verify the de novo synthesis model, cells expressing GFP-lifeact were wounded in the presence of latrunculin A and/or jasplakinolide, a stabilizer of actin filaments (Figure 2A). As described above, GFP-lifeact did not accumulate at the wound site in the presence of latrunculin A (LatA in Figure 1G and Figure 2A). In the presence of jasplakinolide, a large actin aggregate formed inside the cell 30 min after incubation, as previously described (arrowhead, Jasp in Figure 2A) [58]. Upon wounding, actin began to accumulate at the wound site (arrows), but the peak and termination times were significantly retarded (Figure 2B,C). Presumably, jasplakinolide shifts the monomer–polymer equilibrium toward the polymer, resulting in a reduction in the number of actin monomers in the cytosol. However, the reduction may be limited, affecting only the assembly rate, and still enabling actin accumulation. Incidentally, in the presence of jasplakinolide, the PI influx ceased after wounding (Figure 2D).

Next, when cells were incubated with jasplakinolide for 30 min and then with latrunculin A, large and small actin aggregates were observed. Under these conditions, the number of polymerizable actin monomers should be significantly reduced, although the pre-built actin structures such as the aggregates are stabilized by jasplakinolide. Upon wounding, actin did not accumulate at the wound site (Jasp+LatA, Figure 2E), suggesting that actin could not accumulate because it could not newly polymerize without polymerizable actin monomers.

Therefore, it is plausible that actin polymerizes de novo at the wound site.

### 3.4. Early Signals for Wound Response

We have shown that the entry of Ca^2+^ from the external medium through wound pores triggers actin accumulation. The presence of EGTA, a chelator of Ca^2+^, in the external medium completely inhibits actin accumulation [16]. To examine whether other soluble secondary messengers, such as cAMP and cGMP, act as regulators of actin dynamics, cells expressing Flamindo2, a cAMP sensor, or Dd-Green cGull, a cGMP sensor [59], were wounded by laserporation. As a reference, intracellular Ca^2+^ (Ca_i_^2+^) was monitored using cells expressing GCaMP6s, a Ca^2+^ sensor. Neither showed any positive response, although Ca^2+^ levels transiently increased in the cytosol (Appendix A).

Annexins and ESCRT, which are membrane remodeling proteins, have been implicated in cell membrane wound repair [28,29,60,61,62]. We have shown that these proteins immediately accumulate at the wound site upon wounding [13,16]. We examined whether these proteins contribute to wound-induced actin accumulation. Figure 2F and Appendix A show the time courses of fluorescence intensities of annexin C1-null cells and TSG101 (a component of the ESCRT complex)-null cells expressing GFP-lifeact, respectively. The curves often showed multiple peaks for annexin C1-null cells, although TSG101-null cells showed a time course similar to that of the wild-type cells. Therefore, annexin C1 should be located upstream of actin regulation.

We have shown that calmodulin also immediately accumulates at the wound site. W7, a calmodulin inhibitor, inhibits both calmodulin accumulation and wound repair. In addition, W7 inhibits the accumulation of both actin and annexin C1 [16]. Figure 2G shows that W7 significantly inhibited actin accumulation upon wounding when using the improved method, which confirms the previous observation.

Altogether, calmodulin should be placed upstream of annexin C1 and actin.

### 3.5. Dynamics of Actin-Related Proteins during Wound Repair

To build actin-containing structures at the wound site, actin-related proteins (ARPs) participate in proper temporal and spatial regulation. To clarify the types of ARPs that contribute to actin remodeling, cells expressing various GFP-tagged ARPs were wounded. Many types of ARPs transiently accumulated at the wound site at different times (Figure 3A). Appendix A shows descriptions of individual proteins and a summary of their responses, indicating that 13 of the 22 newly examined ARPs accumulated at the wound site. Figure 3B,C show the time courses of the fluorescence intensities of the accumulated ARPs and their normalized curves when the individual peak value was set to 100%. Figure 3D shows the durations of their appearance. Actual time courses of the fluorescence intensities of all the examined GFP-tagged proteins are shown in Appendix A.

Remarkably, severin (an F-actin-severing protein similar to gelsolin) accumulated at the wound site almost immediately. WASP (a protein that initiates actin polymerization), Arp3 (a component of the Arp2/3 complex that nucleates actin filaments activated with WASP), ABP34 (an actin-bundling protein), and MyoB (a type I myosin) accumulated significantly earlier than actin. Alpha-actinin (an actin-crosslinking protein), filamin (an actin-crosslinking protein), and CAP32 (an actin capping protein) began to accumulate simultaneously with actin. MyoC (a type I myosin), CARMIL (a multidomain scaffold protein), and fimbrin (an actin-bundling protein) began to accumulate after actin. Coronin and cofilin accumulated in the actin disassembly stage, consistent with the previous consensus that coronin inactivates the Arp2/3 complex [63] and that ADF/cofilin promotes actin disassembly [64].

Interestingly, EfaA1 (elongation factor 1 alpha 1, actin-bundling protein) transiently disappeared from the wound site (orange bars in Figure 3D and Appendix A). Incidentally, we have reported that myosin II, myosin heavy chain kinase C, PakA, PTEN, and cortexillins A and B transiently disappear from the wound site [27].

Among the examined ARPs, MyoA (a type 1 myosin), SCAR (a member of the WASP family), vinculin (involved in cell–substrate adhesion by interacting with actin), GmfA (an inactivator of Arp2/3), paxillin (involved in cell–substrate adhesion by interacting with actin), formin A (an actin-nucleation protein), and DwwA (a ww-domain-containing protein) did not accumulate at the wound site (Appendix A).

In summary, specific types of ARPs may accumulate with their unique temporal and spatial regulations to remodel actin structures at wound sites.

### 3.6. ARPs Required for Wound Repair

To examine the ARPs required for wound-induced actin dynamics, knockout mutants of these proteins were transformed with GFP-lifeact, and the response of actin accumulation upon wounding was examined. The mutants available in DictyBase and NBRP Nenkin stock centers were mainly used. Several inhibitors were also used. Figure 4 and Figure 5 show the durations and amplitudes (at the peaks) of actin accumulation in the knockout mutants and wild-type cells in the presence of inhibitors upon wounding (red bars), respectively. The actual time courses of the fluorescence intensities are shown in Appendix A.

Although severin immediately accumulated at the wound site as described above, actin normally accumulated at the wound site in severin-null cells (Appendix A). WASP-null cells reduced the amplitude of actin accumulation and retarded the initiation of the accumulation, which was also confirmed by similar results when wild-type cells were wounded in the presence of a WASP inhibitor (wiscostatin) (Appendix A). Inhibitors of Arp2/3 (CK666 or CK869) significantly retarded both the initiation and termination times of actin accumulation and reduced its amplitude (Appendix A). An inhibitor of formin (SMIFH2) also had a similar effect (Appendix A). Coronin-null cells showed significantly retarded termination time, suggesting that coronin contributes to the disassembly of the actin structure (Appendix A).

We have shown that myosin II transiently disappears from the wound site [27]. Nevertheless, myosin II-null and myosin II heavy chain kinase C (MHCKC)-null cells showed actin dynamics comparable to that of wild-type cells (Appendix A).

Profilin A/B double-null cells significantly reduced the amplitude of actin accumulation and retarded termination (Appendix A). These monomer-sequestering proteins contribute to wound-induced actin dynamics.

Other knockout mutants, including filamin-null, α-actinin/filamin double-null, WASH-null, and cortexillin A/B double-null cells, showed responses similar to those of wild-type cells (Appendix A).

Overall, these results indicate that WASP, Arp2/3, formin, profilin, and coronin are required for proper actin dynamics during wound repair in *Dictyostelium* cells.

### 3.7. Signals for Wound-Induced Actin Dynamics

Signal cascades for chemotaxis in *Dictyostelium* as well as neutrophils have been intensively investigated [65,66,67,68]. Multiple signaling pathways that regulate actin remodeling have been implicated: cGMP, PI3K, PLA2, the Elmo/Dock complex, and TORC2 pathways. Some of these pathways might be associated with wound-induced actin dynamics. Figure 4 and Figure 5 show the durations and amplitudes (at the peaks) of actin accumulation in the signal-related protein knockout mutants and wild-type cells in the presence of specific inhibitors upon wounding (purple). The actual time courses of the fluorescence intensities are shown in Appendix A

The influx of Ca^2+^ induces Ca^2+^ release from the endoplasmic reticulum, mediated by inositol 1,4,5-triphosphate (IP3) receptor-like protein A (IplA), which is homologous to IP3 receptors [69]. Wounded IplA-null cells show much lower Ca_i_^2+^ responses than wild-type cells [13]. As expected, actin dynamics in the mutant cells showed lower amplitudes and retarded termination (Appendix A).

Double-knockout mutants of GcA/SgcA (membrane-bound and soluble-guanylyl cyclases) and GbpC/GbpD (cGMP-binding proteins) showed an actin accumulation curve similar to that of wild-type cells (Appendix A), indicating that the cGMP pathway is not involved in wound-induced actin accumulation. Knockout mutants of adenylyl cyclase (AcA) also showed actin accumulation similar to that in wild-type cells (Appendix A), indicating that the cAMP pathway is not involved in actin accumulation.

PI3K is a key regulator of actin polymerization in chemotactic responses [70,71]. The GFP-Pleckstrin-homology (PH) domain of PKB, a marker of PIP3, is enriched at the leading edge of chemotactic cells [70]. However, the GFP-PH domain did not accumulate at the wound site (Appendix A), and actin almost normally accumulated at the wound site in PI3K quintuple-null cells or in the presence of a PI3K inhibitor (LY294002) (Appendix A).

In contrast, GFP-PTEN (phosphatase and tensin homolog deleted from chromosome 10), which converts PIP3 to phosphatidylinositol 3,4-bisphosphate (PIP2), transiently disappeared from the wound site [27], and the termination of actin dynamics was considerably retarded in PTEN-null cells (Appendix A). In addition, an inhibitor of phospholipase C (PLC), which cleaves PIP2, also increased the termination of actin accumulation (Appendix A). Therefore, the PI3K pathway in chemotaxis does not involve wound-induced actin accumulation, but PIP2-derived products appear to mediate the termination of actin dynamics via PTEN and PLC.

PLA2 (phospholipase A2) is one of the signaling proteins involved in chemotaxis, and acts parallel to the PI3K pathway [72,73]. A PLA2 inhibitor (*p*-bromophenacyl bromide, BPB) significantly retarded the peak and termination times of actin dynamics (Appendix A). The Elmo/Dock complex interacts with the Gβ subunit to transduce the GPCR signal to the actin cytoskeleton by regulating small G proteins such as Rac [68]. Knockout of ElmoE significantly reduced the amplitude and retarded the termination time (Appendix A), suggesting that the Elmo/Dock complex pathway mediates wound-induced actin dynamics.

The TORC2 also plays a critical role in chemotaxis [74]. Activated RasC stimulates TORC2 to phosphorylate the downstream protein kinases, PKB and PKBR1, as well as AcA [75,76]. An inhibitor of TORC2 (Torin1) retarded both the initiation and termination of actin accumulation (Appendix A). PKB, which phosphorylates many different signal proteins such as PakA and RasC to regulate actin polymerization at the leading edge, is regulated by TORC2 [74,76,77]. However, PKB-null cells showed wound-induced actin responses that were comparable to those of wild-type cells (Appendix A). PKC (protein kinase C) has been reported to be involved in wound repair in other organisms [12,37,78,79]. PkcA, a PKC ortholog in *Dictyostelium*, is involved in chemotaxis [80,81]. An inhibitor of PKC (bisindolylmaleimide 1, BIM1) did not affect wound-induced actin dynamics (Appendix A).

Overall, the pathways involving PIP2-derived products, PLA2, the Elmo/Dock complex, and TORC2 mediate wound-induced actin dynamics.

### 3.8. Small G Proteins for Wound-Induced Actin Dynamics

The Rho family of small G proteins is an upstream signal for regulating actin dynamics during chemotaxis and may be shared with wound-induced actin dynamics. *Dictyostelium* cells have 20 Racs but no Rho and Cdc42 [66]. A Rac inhibitor (EHT1864) significantly reduced the amplitude of actin accumulation and retarded both the initiation and termination times (Appendix A), suggesting that Rac regulates wound-induced actin dynamics. In this study, we only examined RacA (Appendix A) and RacE (Appendix A) but they did not seem to participate in wound-induced actin dynamics. Other members of Racs may mediate actin dynamics.

RasG and RasC, members of the Ras family of small G proteins, are critical mediators of chemotactic responses [82]. RasG activates PI3K and regulates actin polymerization and the direction of cell migration. RasC also stimulates TORC2 to phosphorylate PKB and PkbR1, as well as to activate AcA [75]. The deletion of both RasG and RasC did not affect wound-induced actin accumulation (Appendix A).

### 3.9. Other Signals for Wound-Induced Actin Dynamics

P21-activated protein kinases (Paks) act on a large number of regulators of the actin cytoskeleton in higher eukaryotes, including RhoGDI, LIM kinase, filamin, diverse myosin kinases, the Arp2/3 complex, and paxillin [66]. However, the deletion of PakA, one of the at least nine Paks in *Dictyostelium*, did not affect actin accumulation (Appendix A). The ortholog of Huntington’s disease protein Huntingtin (Htt) regulates myosin II phosphorylation through phosphatase PP2A, affecting chemotaxis and cytokinesis [83,84]. The deletion of Htt did not affect wound-induced actin accumulation (Appendix A).

The serine/threonine kinase, Phg2, binds Ras GTPases and PIP2 to regulate phagocytosis, macropinocytosis, cell–substrate adhesion, cytokinesis, and actin filament organization [85,86]. The deletion of Phg2 did not affect wound-induced actin accumulation (Appendix A). Tortoise, a novel protein with a predicted coiled-coil motif, is required for chemotaxis [87]. The deletion of Tortoise A (TorA) significantly reduced wound-induced actin accumulation (Appendix A). Tortoise has not been reported to be directly associated with the actin cytoskeleton.

Cyclase-associated protein (CAP) is an important regulator of cell polarity, chemotaxis, and cell adhesion, and is linked to the actin cytoskeleton [88]. GFP-CAP did not accumulate at the wound site (Appendix A). Stress-activated protein kinase A (SpkA) is involved in chemotaxis, cytokinesis, and actin regulation [89]. GFP-SpkA did not accumulate at the wound site (Appendix A).

Microtubules have been implicated in the assembly of actin structures and transfer of membrane vesicles for wound repair in higher organisms [35,90,91]. Actin accumulated normally in the presence of thiabendazole (TB), a microtubule depolymerizer (Appendix A). In addition, deletion of Kif12, one of the 13 kinesins (microtubule-dependent motor proteins) in *Dictyostelium*, did not affect actin accumulation (Appendix A). Our previous PI influx experiments also showed that wounds normally close in the presence of TB [16], suggesting that microtubules do not seem to be required for wound repair.

Upon wounding, a membrane plug immediately forms to close the wound pore [16]. In addition to the accumulation of annexin C1 and ESCRT components, we have reported that clathrin and a dynamin-like protein (DlpA), both of which are membrane remodeling proteins, transiently disappear from the wound site [27]. Although clathrin-null cells showed actin accumulation comparable to that in wild-type cells (Appendix A), DlpA-null cells showed significantly reduced amplitude of actin dynamics (Appendix A). Incidentally, DlpA has been reported to stabilize actin filaments [92,93]. We examined other dynamins, including DlpB and DymA. GFP-DymA did not accumulate at the wound site, and deletion of DlpB resulted in almost normal actin accumulation (Appendix A, respectively).

Interestingly, deletion of LvsA, which is also a membrane remodeling protein [94], significantly reduced the amplitude of actin dynamics (Appendix A). Incidentally, LvsA also appears to be involved in Ca^2+^ mobilization during chemotaxis [69].

The responses of the examined proteins during wound repair, including our previously reported data, are summarized in Appendix A.

## 4. Discussion

We improved our laserporation method by replacing the carbon with gold coating, which greatly improved the fluorescence images. The energy absorbed in gold may generate heat and plasmons [48], which wounds only the cell membrane. *Dictyostelium* cells live in soil in nature and are cultured in plastic dishes in laboratory. Gold is inert and has no affects to cells [95]. We believe that this method is applicable to cells of other organisms.

By using this method, we confirmed that actin accumulation at wound sites is essential for wound repair. We also found that actin assembles by de novo synthesis in *Dictyostelium* cells, although it has been reported to accumulate by myosin II-dependent cortical flow with dynamic assembly in large eggs [57]. We have shown that myosin II-null cells exhibited no substantial defect during wound repair in PI influx experiments [27], and in the present study, they showed no substantial defect of actin accumulation. Although constriction of the actomyosin ring contributes to cellular wound closure in eggs and embryos, it is not essential for wound repair in smaller cells. We also found that MyoB and MyoC accumulate at the wound site. Since MyoB begins to accumulate earlier than actin, it may anchor actin filaments to the cell membrane, as previously described [96]. Experiments using knockout mutants are required to determine whether it contributes to actin accumulation.

PI influx experiments showed that the wound substantially closes within 2–3 s, and that actin transiently accumulates between 2.5 and 24 s, with a peak at 6 s, after wounding. We have proposed a two-step model for wound repair [16]. In this model, upon wounding, Ca^2+^ influx triggers the de novo generation of vesicles and the mutual fusion of vesicle–vesicle and vesicle–cell membranes to create an urgent membrane plug. Actin then accumulated to complete the plug. In the future, the exact role of actin accumulation at the molecular level should be elucidated.

When *Dictyostelium* cells are stimulated with a chemoattractant, actin accumulation begins in the cell cortex after 3–5 s and terminates at 15 s (the first peak), followed by the second peak (25–40 s) [97,98,99]. This discrepancy in the time courses between chemoattractant- and wound-induced responses may be caused by differences in participating ARPs or signals. However, we found that there were many common ARPs and signals in chemotaxis and wound responses.

Our experiments using GFP-tagged proteins showed that specific types of ARPs accumulate at wound sites with different times to remodel actin structures. Surprisingly, severin immediately accumulates at the wound site. Since severin is a Ca^2+^-dependent actin-severing protein, we first hypothesized that it may cut and remove actin filaments that are damaged or disordered by wounding. However, we did not observe any removal of cortical actin filaments in wild-type cells, and the actin dynamics in severin-null cells were similar to those in wild-type cells.

WASP, Arp2/3, formin, and ABP34 accumulated earlier at the wound site than actin. Although we did not examine the effects of the inhibition of ABP34, the inhibition of WASP, Arp2/3, or formin retarded both the initiation and termination times of actin accumulation and reduced the amplitude, suggesting that these proteins contribute to actin assembly at the wound site. Although the Arp2/3 complex and formins have been considered to play different roles in actin regulation, their regulation may be partially synergistic [100]. The formin family comprises 10 different and functionally distinct proteins in *Dictyostelium* [66,101,102]. Exact type of formin responsible for wound repair should be clarified in future studies.

Several ABPs, such as α-actinin, fimbrin, severin, and ABP34, which are regulated by Ca^2+^ in vitro, accumulate at the wound site. However, knockout mutants (ABP34-null was not determined) did not affect actin accumulation. Incidentally, single-knockout mutants of these ABPs show no distinct phenotypes of cell migration, chemotaxis, or cell division in *Dictyostelium* [66].

Coronin and cofilin accumulate at the wound site in the stage of actin disassembly. Although we could not inhibit cofilin because it is essential and null mutants are not available [103], coronin-null cells significantly retarded the termination of actin dynamics, suggesting that coronin and probably cofilin contribute to actin disassembly at the wound site. Incidentally, coronin enhances cofilin-mediated severing by accelerating cofilin binding to actin filament sides [104]. One of the reasons why most ARP knockout cells did not affect wound-induced actin accumulation may be their overlapping functions, as previously discussed [105].

Chemotaxis signaling pathways were used as a reference. Among these, the cAMP and cGMP pathways did not mediate wound-induced actin accumulation. PI3K pathways did not mediate actin accumulation either, although PI3K signaling via RasG/C and PKB is one of the key regulators of actin dynamics for chemotaxis [106]. Independent of PI3K, PIP2-derived products may mediate wound-induced actin dynamics because PTEN and PLC null cells elongate the termination of actin dynamics. Incidentally, PIP2 itself is also implicated in regulating actin remodeling [107,108], but it should be excluded as a signal because GFP-PTEN, which is also a marker of PIP2, did not accumulate at the wound site.

The PLA2 pathway is also a candidate, because this specific inhibitor retarded the termination of actin accumulation. PLA2 generates arachidonic acid-related lipids in an intracellular Ca^2+^-dependent manner. The PLA2 pathway is involved in chemotaxis in parallel to the PI3K pathway [72]. In addition, PLA2 contributes to increase the levels of Ca_i_^2+^ [109]. The Elmo/Dock complex and TORC2 pathways mediate wound-induced actin accumulation based on the results of inhibition experiments. The inhibition of these signals significantly retarded both the initiation and termination times of actin dynamics, indicating that they mediate actin dynamics.

The Rac inhibitor retarded both the initiation and termination times of actin dynamics and reduced the amplitude, suggesting that a member of the Rac family contributes to actin accumulation. Presumably, such a Rac may be activated by RacGEF and then activates WASP, Arp2/3, and formin to induce actin assembly. Since *Dictyostelium* cells have 20 Racs, the exact Rac responsible for wound-induced actin accumulation needs to be identified in the future.

In the present study, we found that calmodulin is located upstream of annexin C1 and actin. The counterpart of calmodulin is yet to be identified. Incidentally, there are many calmodulin-binding proteins, including those related to actin regulation, such as DwwA, Htt, RgaA, MHCKs A and B, Paks A and B, GapA, RacGEF, CaM-BP46, and MyoG [110]. The connection between calmodulin and annexin C1 remains to be clarified. In addition, the connections between calmodulin and newly identified pathways such as the TORC2, Dock/Elmo, PIP2-derived product, and PLA2 pathways are yet to be elucidated.

We have summarized the signaling pathways for actin dynamics in wound repair in Figure 6, in comparison to those of chemotaxis. After Ca^2+^ enters the cytosol upon wounding, calmodulin and annexin C1 accumulate immediately at the wound site, which finally induces actin assembly. The TORC2, Dock/Elmo, PIP2-derived product, and PLA2 pathways are common in chemotaxis and wound repair. Racs, WASPs, and then formins and Arp2/3 are involved in these pathways, and further downstream, many ARPs regulate the actin dynamics at the wound site. We believe that these pathways can be completed by combining multiple gene-knockout mutants and inhibitors in the future. Since the induction of local actin assembly by wounding is highly reproducible, the laserporation method is a powerful tool to obtain general and useful information on actin dynamics during other actin-related phenomena such as chemotaxis, phagocytosis, and cell–substrate adhesion.

## Figures and Tables

**Figure 1 cells-11-03166-f001:**
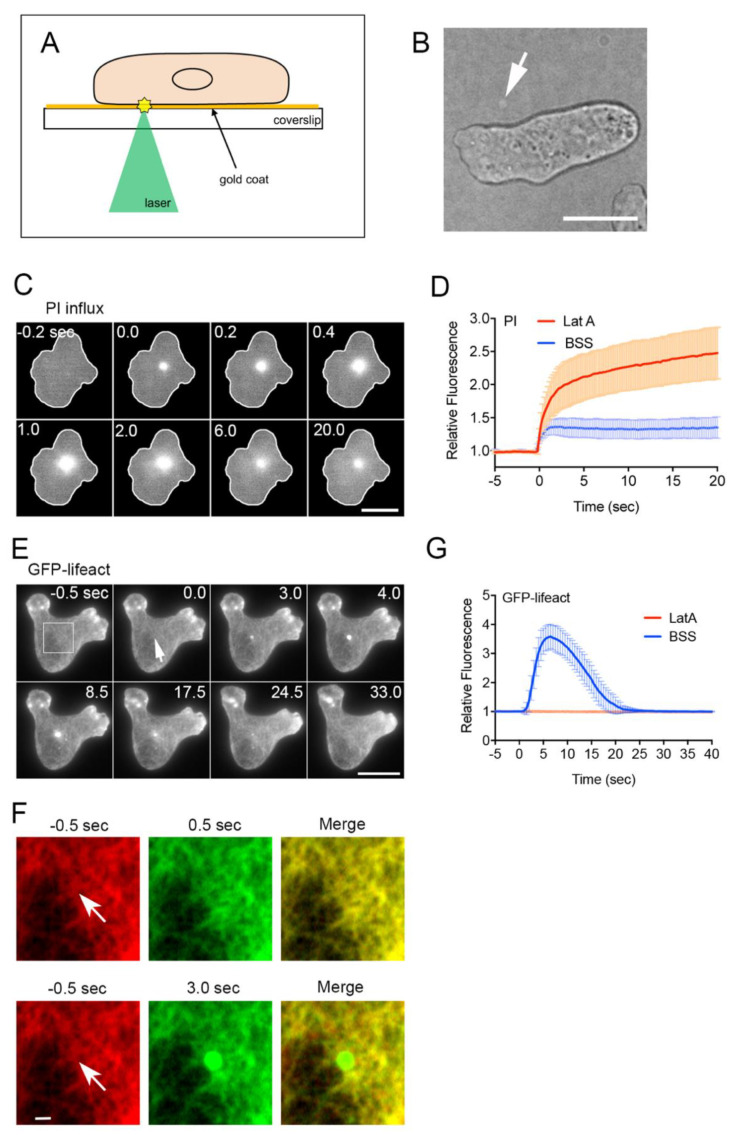
Actin accumulation at wound sites is essential for wound repair. (**A**) Schema for the improved laserporation using gold coating. The wound diameter was set as 0.5 µm. (**B**) When the laser beam was focused on the surface of the coated coverslip, the gold coating peeled off, leaving behind a small white spot (arrow). (**C**) A typical sequence of fluorescence images of PI influx after laserporation. The wound laser beam was applied at 0 s and the duration was set at 8 ms. (**D**) Time courses of PI influx in the presence (LatA) and absence (BSS) of latrunculin A (*n* = 25 each). (**E**) A typical sequence of fluorescence images of a cell expressing GFP-lifeact after laserporation. The arrow shows the wound site. (**F**) Enlarged images of the rectangle in panel E 0.5 s before wounding (−0.5 s, red) and 0.5 or 3.0 s after wounding (0.5 s in upper panel and 3.0 s in lower panel, green), and the merged images. Arrows show the wound sites. (**G**) Time courses of fluorescence intensities at the wound site in the presence (LatA) and absence (BSS) of latrunculin A. The fluorescence intensities were normalized by setting the value before wounding to 1 after subtracting the background (*n* = 25 each). Scale bars: 10 µm for panels B, C, and D, and 1 µm for panel F.

**Figure 2 cells-11-03166-f002:**
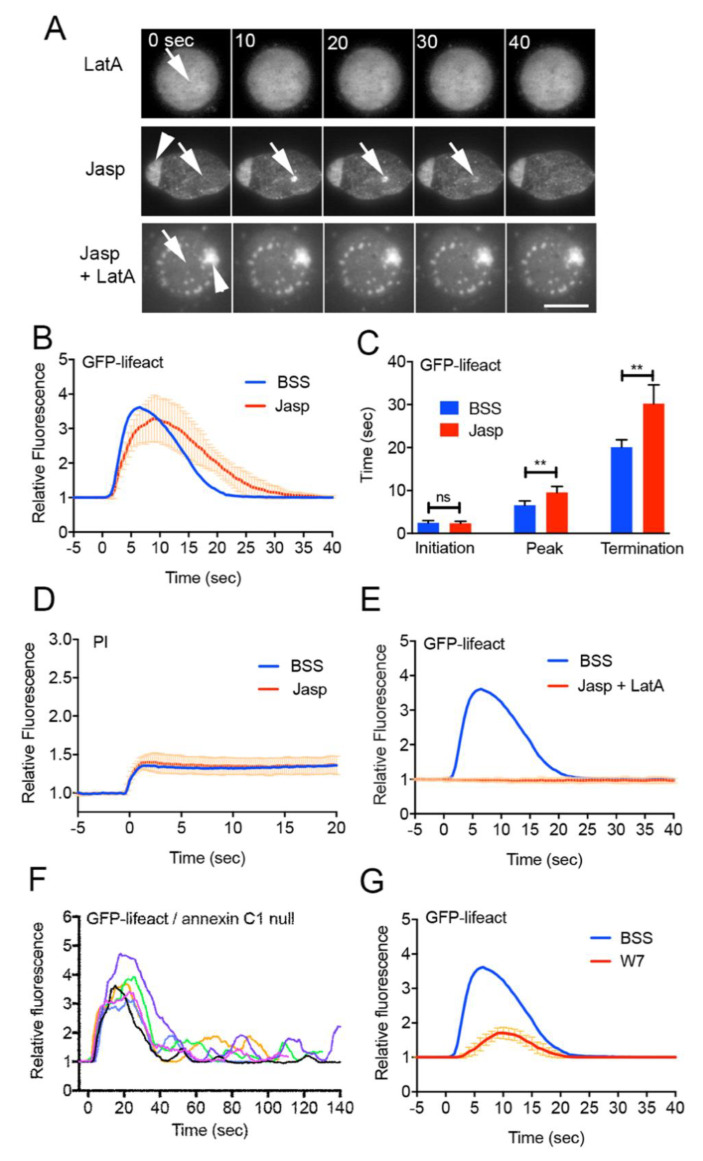
Actin polymerizes de novo at wound sites. (**A**) Typical sequences of fluorescence images of cells expressing GFP-lifeact in the presence of latrunculin A and/or jasplakinolide. Arrows and arrowheads show the wound sites and large actin aggregates, respectively. (**B**) Time courses of fluorescence intensities of GFP-lifeact at the wound site in the presence (Jasp) and absence (BSS) of jasplakinolide, respectively. Note that the time courses in the presence of latrunculin A are shown in Figure 1G. (**C**) The initiation, peak, and termination times of wound-induced actin accumulation are compared in the presence and absence of jasplakinolide (*n* = 25 each). Data are presented as means ± SD. ** *p ≤* 0.0001; *ns*: not significant, *p* > 0.05. (**D**) Time courses of PI influx in the presence and absence of jasplakinolide (*n* = 25 each). (**E**) Time courses of fluorescence intensities of GFP-lifeact at the wound site in the presence and absence of both inhibitors (Jasp + LatA) (*n* = 25 each). (**F**) Time courses of fluorescence intensities of GFP-lifeact at the wound site in annexin C1-null cells. The graphs with different colors show time courses of six different cells. (**G**) Time courses of fluorescence intensities of GFP-lifeact at the wound site in wild-type cells in the presence and absence of W7 (*n* = 25 each).

**Figure 3 cells-11-03166-f003:**
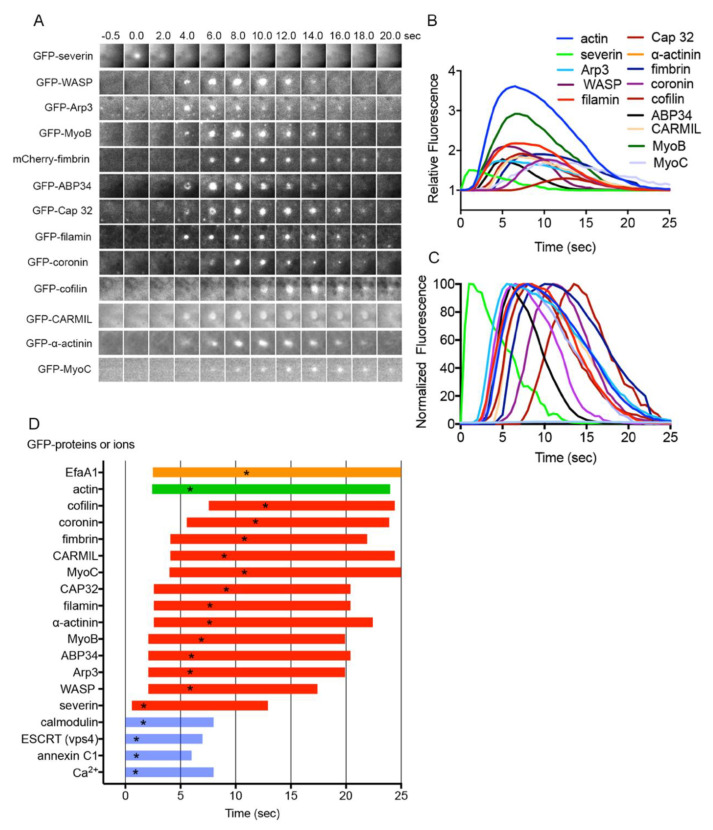
Dynamics of ARPs during wound repair. (**A**) Typical sequences of fluorescence images of cells expressing various (13 kinds) GFP-ARPs at wound sites. (**B**) Time courses of fluorescence intensities of GFP-ARPs and GFP-actin at wound sites (*n* = 25 each). (**C**) Normalized curves of panel B when each peak value is set as 100%. (**D**) Duration of appearance of individual ARPs at wound sites (red). Duration of actin dynamics (green), Ca^2+^ influx (blue), calmodulin dynamics (blue), ESCRT component vps4 dynamics (blue), and annexin C1 dynamics (blue) are also plotted. In addition, the duration of Efa1A disappearance (orange) is plotted. Asterisks in the duration bars indicate the peak times.

**Figure 4 cells-11-03166-f004:**
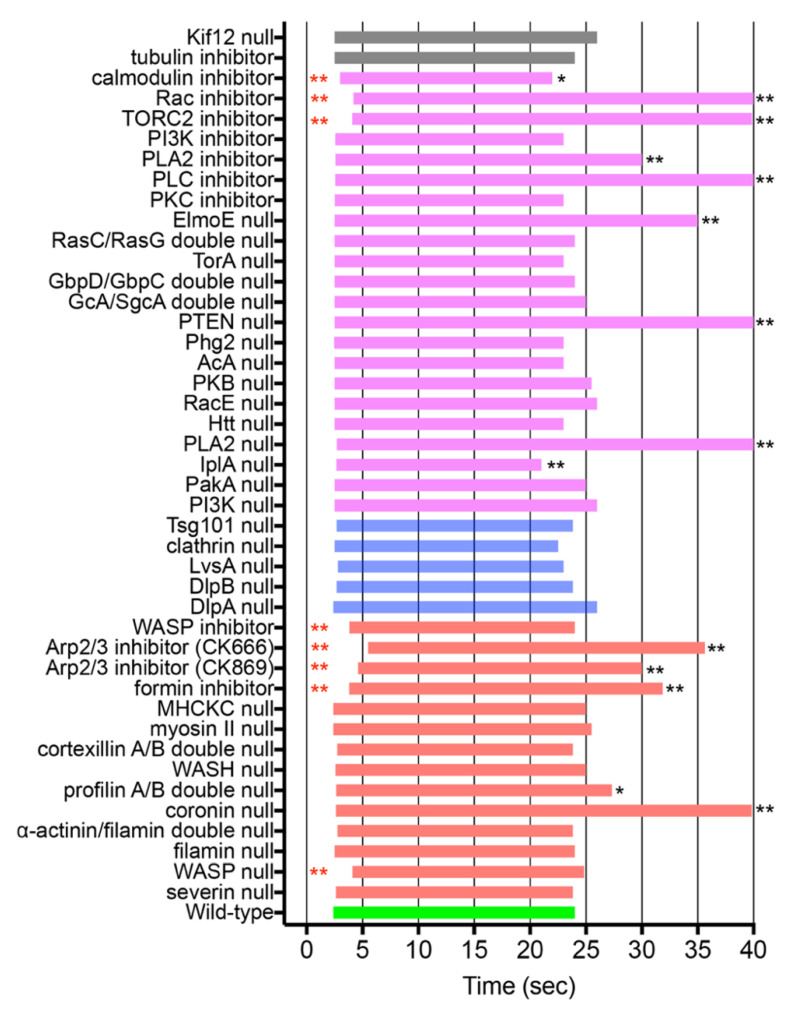
Durations of wound-induced actin dynamics in null cells or wild-type cells in the presence of inhibitors. Durations of wound-induced actin dynamics in null cells or wild-type cells in the presence of inhibitors: actin-related (red), signal-related (purple), membrane dynamics-related (blue), and microtubule-related (gray) proteins. Actual time courses of the fluorescence intensities are shown in Appendix A. Red asterisks indicate the cases showing significantly retarded initiation time compared with the control, and black asterisks indicate the cases showing significantly retarded termination time. * *p* ≤ 0.001, ** *p* ≤ 0.0001 (*n* ≥ 25 each).

**Figure 5 cells-11-03166-f005:**
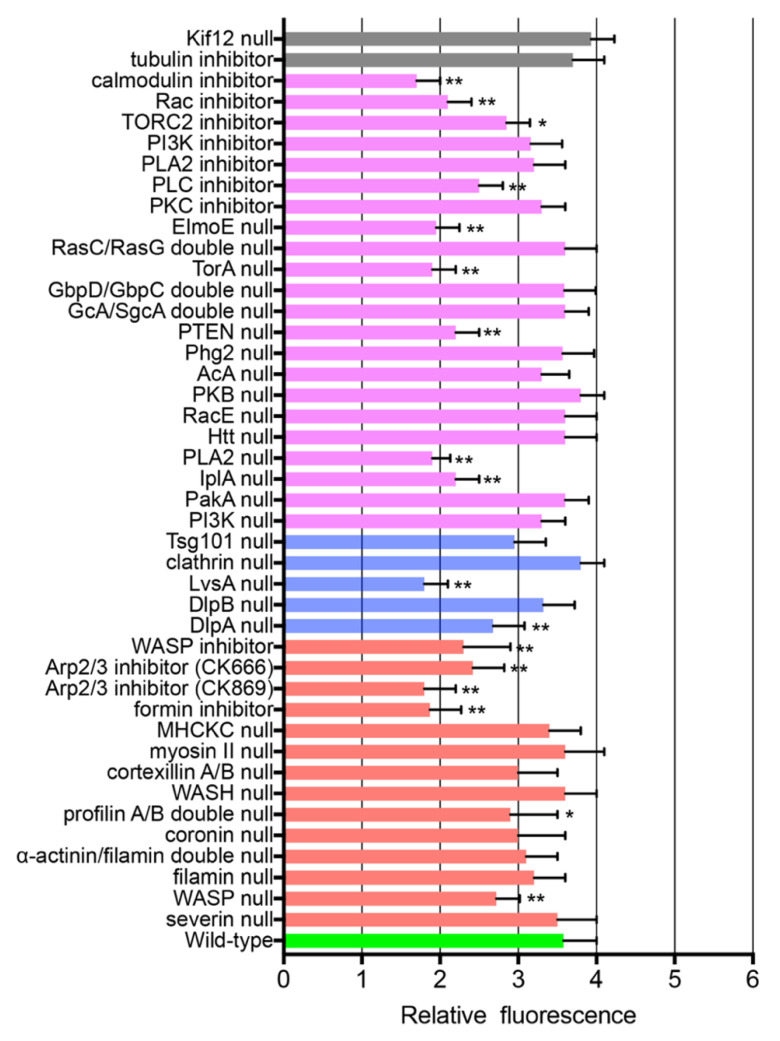
Peak amplitudes of wound-induced actin dynamics in null cells or wild-type cells in the presence of inhibitors. The amplitudes at the peaks of the actin dynamics under the same conditions as shown in Figure 4. Actual time courses of the fluorescence intensities are shown in Appendix A. Asterisks show cases with significant differences from wild-type cells in the absence of inhibitors. Data are presented as means ± SD. * *p* ≤ 0.001, ** *p* ≤ 0.0001 (*n* ≥ 25 each).

**Figure 6 cells-11-03166-f006:**
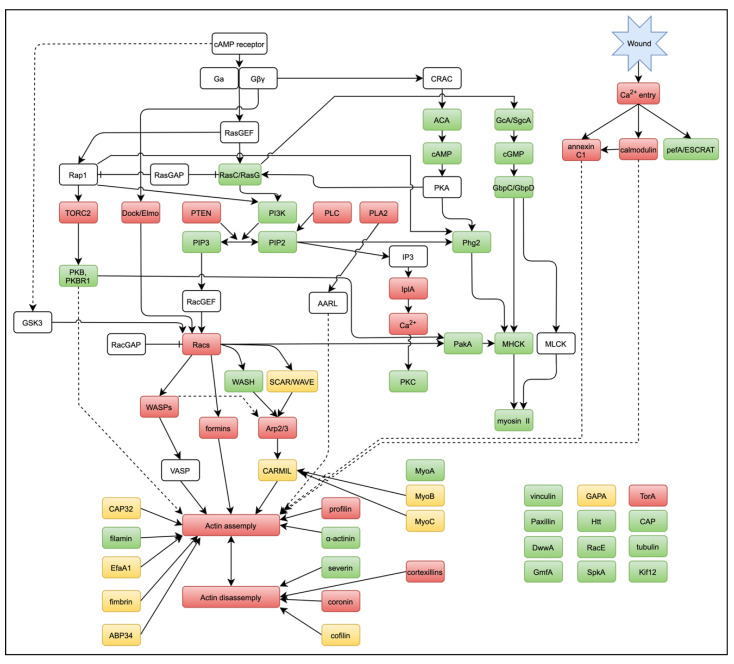
Signal pathways for actin dynamics in wound repair. With reference to the signal cascades for chemotaxis, potent candidates for wound-related signals, ARPs, or other proteins we found in the present and previous studies are marked (red). Some proteins (yellow) accumulate or disappear at the wound site, but their contribution remains to be clarified using knockout or inhibitor experiments. Most of the examined proteins (green) do not contribute to wound repair.

## Data Availability

All relevant data are available from the authors on reasonable request.

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
