# Peer review of "Dynamics of Actin Cytoskeleton and Their Signaling Pathways during Cellular Wound Repair"

_cells, 2022, doi:10.3390/cells11193166_

Round 1
Reviewer 1 Report
By using TIRF microscopy and laserporation in Dictyostelium cells, the authors study F-actin accumulation and recruitment of actin-related proteins at wound sites. They further disrupted the expression or activity of a set of different signaling proteins and analyzed their effects on the transient accumulation of F-actin. With this strategy they identified the temporal contribution of different signaling molecules and proposed a signaling pathway for F-action formation at wound sites.
This is a very good work that importantly contributes to get a better comprehension of the mechanisms involved in wound repair.
I only have minor comments:
1.- The study model should be mentioned in the abstract.
2.- Some details are omitted in some figure legends, for example, in Figure 1D and 1G is not indicated the number of experiments, and if the bars indicate SD or SE, and in Figure F it is not indicated what each color represents. Such type of information should be included.
Author Response
First of all, we would like to greatly appreciate the reviewer’s helpful comments.
C1. The study model should be mentioned in the abstract.
A1. We added model name ‘Dictyostelium’ in the abstract.
C2. Some details are omitted in some figure legends, for example, in Figure 1D and 1G is not indicated the number of experiments, and if the bars indicate SD or SE, and in Figure F it is not indicated what each color represents. Such type of information should be included.
A2. We added the number of experiments. We also added explanation of each color.
Reviewer 2 Report
I reviewed the paper entitled: “Dynamics of actin cytoskeleton and their signaling pathways 2 during cellular wound repair”, which presents an improved method for laserporation in order to study membrane wound healing. They show consistency with the author´s previous work and added new valuable data. I believe the manuscript is publishable after improving mainly format issues, that I describe below:
Within the manuscript:
Every time they say wound closure it should be preceded by cellular, for instance lanes 85, 87, 91. Review the complete manuscript.
Introduction
Lanes 31 32 . Rephrasing since it is not accurate to say that the loss of wound closure causes muscular dystrophy (a genetic disease) instead loss of cellular wound closure is seen in diseases like MD.
Lane 56 sentence finished with “by cutting” this seam as an unfinished sentence.
Methods
Lanes 104-105 “the vectors were transformed” should be changed to the cells were transformed with the vectors.
Lane 124 first time TIRF is mentioned, then it should say Total Internal Reflection Fluorescence (TIRF) microscopy
Lanes 133 – 134 The title should say Propidium Iodide in full. Within the paragraph propidium iodide (PI)
Explain the concentration of several inhibitors used. They were used at their IC50 , below, or 2X IC50.
Results
161 use abbreviature TIRF.
175 use PI
Explain the rationale for the use of 3mM Ca+2 to the PI experiment. This should have been explained in Methods.
Fig.1 F The legend is not self-explanatory. State the meaning of green and red.
An experiment showing the microscopy observation that Lat incubation let to a persistence of the pore. This will make the conclusion affirming that actin polymerization is essential for cell wound repair more robust.
Lane 236-237, legend 2B. Please check, since the figure only has JASP results.
It is nice that the main conclusion of each experiment is included at the end of each series, however, some are to absolute. Revise lane 250, it cannot totally rule out that cortical actin might, in certain extension move to the wounded area. Since JASP is a stabilizer, the author state that this will give less monomers availability for de novo polymerization, however it is not demonstrated with the presented results.
Lane 242, figure legend for 2F does not explain different colors for the curves on the graph (blue, black, yellow, green, etcetera)
Lane 348 it should say: cellular wound repair…add “in Dictyostellum”
Lane 359: To the figure tittle add: “transformed” or wild type cells .
Lane 418 check why it is written in italics
Lanes 526-527 consider rephrasing, it is confusing
Lane 542 change this for “its”
Lane 550 delete “an”
References
Double check if there are more current references for the topic
Conclusion
Missing
Author Response
First of all, we would like to greatly appreciate the reviewer’s helpful comments.
C1. Every time they say wound closure it should be preceded by cellular, for instance lanes 85, 87, 91. Review the complete manuscript.
A2. We would like to change ‘wound closure’ to ‘cellular wound closure’ or ‘wound repair’
C2. Lanes 31 32 . Rephrasing since it is not accurate to say that the loss of wound closure causes muscular dystrophy (a genetic disease) instead loss of cellular wound closure is seen in diseases like MD.
A2. We changed the phrase as the reviewer suggested.
C3. Lane 56 sentence finished with “by cutting” this seam as an unfinished sentence.
A3. We changed the phrase, as follows.
‘removal of the wounded membrane by cutting’
à ‘cutting off the wound membrane’
C4. Lanes 104-105 “the vectors were transformed” should be changed to the cells were transformed with the vectors.
A4. We corrected the mistake.
C5. Lane 124 first time TIRF is mentioned, then it should say Total Internal Reflection Fluorescence (TIRF) microscopy
A5. We corrected the mistake.
C6. Lanes 133 – 134 The title should say Propidium Iodide in full. Within the paragraph propidium iodide (PI)
A6. We added full name.
C7/. Explain the concentration of several inhibitors used. They were used at their IC50 , below, or 2X IC50.
A7. We added the explanations in Methods.
Inhibitors were used at a higher than IC50 concentration. We add more details of inhibitors as follows.
Inhibitors were used at a higher than IC50 concentration. The concentrations of CK666, SMIFH2, U731220, BPB, EHT1864, wiscostatin, and torin1 was used based on macropinocytosis inhibition in previous report (Williams and Kay, 2018). The concentrations of latrunculin A, jasplakinolide, W7 hydrochrolide, and thiabendazole were used as previously described (Uddin et al., 2020). The IC50 concentrations of CK869, SMIFH2, EHT1864 were 30, 20, 20 µM, respectively, which were determined by examining cell velocities.
C8. 161 use abbreviature TIRF.
A8. We changed it according to the reviewer.
C9. 175 use PI
A9. We changed it according to the reviewer.
C10. Explain the rationale for the use of 3mM Ca+2 to the PI experiment. This should have been explained in Methods.
A10. BSS regularly contains Ca2+. We would like to remove the phrase ‘containing 3mM Ca2+’.
C11. Fig.1 F The legend is not self-explanatory. State the meaning of green and red.
An experiment showing the microscopy observation that Lat incubation let to a persistence of the pore. This will make the conclusion affirming that actin polymerization is essential for cell wound repair more robust.
A11. We added explanation of green and red.
Regarding the reviewer’ suggested experiments, we previously performed it by staining cell membrane (Uddin et al., 2020). The wound pore does not close in the presence of Lat. We would like to add this information by citing the reference.
Line 220: In addition, we previously reported that the wound hole did not close in the presence of latrunculin A by staining the cell membrane (Uddin et al., 2020).
C12. Lane 236-237, legend 2B. Please check, since the figure only has JASP results.
It is nice that the main conclusion of each experiment is included at the end of each series, however, some are to absolute. Revise lane 250, it cannot totally rule out that cortical actin might, in certain extension move to the wounded area. Since JASP is a stabilizer, the author state that this will give less monomers availability for de novo polymerization, however it is not demonstrated with the presented results.
A12. The data of latrunculin A is already shown in Figure 1G. We added a following sentence in the legend.
Note that the time courses in the presence of latrunculin A are shown in Figure 1G.
Regarding the possibility of the actin flow toward the wound site, the observation by TIRF microscopy (Fig.1F) denies the possibility. However, whether there is no free actin monomer in the presence of both inhibitors is highly plausible but not proved as the reviewer suggested. Therefore, we would like to rephrase this sentence as follows.
Under these conditions, the number of polymerizable actin monomers should be significantly reduced. Actin did not accumulate at the wound site (Jasp+LatA, Fig. 2E), suggesting that it could not accumulate without polymerizable monomers. Therefore, actin polymerizes de novo at the wound site.
à Under these conditions, the number of polymerizable actin monomers should be significantly reduced, although the pre-built actin structures such as the aggregates are stabilized by jasplakinolide. Upon, wounding actin did not accumulate at the wound site (Jasp+LatA, Fig. 2E), suggesting that it could not accumulate because it could not newly polymerize without polymerizable actin monomers.
Therefore, it is plausible that actin polymerizes de novo at the wound site.
C13. Lane 242, figure legend for 2F does not explain different colors for the curves on the graph (blue, black, yellow, green, etcetera)
A13. Each graph is derived from different cells. We would like to add this explanation in the legend.
C14. Lane 348 it should say: cellular wound repair…add “in Dictyostellum”
A14. ‘in Dictyostelium’ was added, according to the reviewer’ suggestion.
C15, Lane 359: To the figure tittle add: “transformed” or wild type cells .
A15. ‘null’ was replaced with ‘cells’.
C16. Lane 418 check why it is written in italics
A16. We fixed the mistake.
C17. Lanes 526-527 consider rephrasing, it is confusingA17. The type of formin responsible for wound-induced actin accumulation should be clarified in future studies. à Exact type of formin responsible for wound repair should be clarified in future studies.
C18. Lane 542 change this for “its”
A18. ‘its’ was changed to ‘the’.
C19. Lane 550 delete “an”
A19. We fixed the mistake.
C20. Double check if there are more current references for the topic
A20. We would like to cite a recent review. Hui et.al., 2022.
Reviewer 3 Report
This paper is a broad survey of a wide variety of different proteins involved in the accumulation of actin at wound sites using the Dictyostelium model. The introduction provides a nice overview of the insights gained from wound repair studies in model organisms and other systems. The results will be interesting to the actin cytoskeleton community and serve as a collection of starting points for deeper analysis of each of the contributing factors.
Comment on how the concentrations of inhibitors were chosen, and whether these levels might be expected to have off-target or non-specific effects in Dicty.
The results and much of the discussion read like a list. This is understandable, since the paper reports the results of a large number of different gene knockouts and inhibitor experiments. The paper is best when the results are synthesized and interpreted in terms of pathways or sets of related proteins. I recommend reorganizing a bit so that short one and two sentence paragraphs are combined into a more holistic description. Fig. 6 and S1 provide an organizing framework that might help.
Minor:
Fig. 1F, please include scale bars.
Line 337-349; if possible organize into a paragraph.
Line 418 odd italics
Fig. S1 Are we sure the cGMP and cAMP sensors are working?
Author Response
First of all, we would like to greatly appreciate the reviewer’s helpful comments.
C1. Comment on how the concentrations of inhibitors were chosen, and whether these levels might be expected to have off-target or non-specific effects in Dicty.
A1. We added the explanations in Methods.
Inhibitors were used at a higher than IC50 concentration. The concentrations of CK666, SMIFH2, U731220, BPB, EHT1864, wiscostatin, and torin1 was used based on macropinocytosis inhibition in previous report (Williams and Kay, 2018). The concentrations of latrunculin A, jasplakinolide, W7 hydrochrolide, and thiabendazole were used as previously described (Uddin et al., 2020). The IC50 concentrations of CK869, SMIFH2, EHT1864 were 30, 20, 20 µM, respectively, which were determined by examining cell velocities.
C2. The results and much of the discussion read like a list. This is understandable, since the paper reports the results of a large number of different gene knockouts and inhibitor experiments. The paper is best when the results are synthesized and interpreted in terms of pathways or sets of related proteins. I recommend reorganizing a bit so that short one and two sentence paragraphs are combined into a more holistic description. Fig. 6 and S1 provide an organizing framework that might help.
A2. As the reviewer’s suggestion, short one and two sentence paragraphs are combined.
Regarding Fig. 6 and S1, we would not like to combine them, because the data in Fig. S1 show negative.
C3. Fig. 1F, please include scale bars.
A3. Scale bar is added in Fig.1F.
C4. Line 337-349; if possible organize into a paragraph.
A4. These lines are organized into a paragraph.
C5. Line 418 odd italics
A5. The mistake is fixed.
C6. Fig. S1 Are we sure the cGMP and cAMP sensors are working?
A6. The cAMP sensor Framindo2 and cGMP sensor Green cGull are already published, which are cited in text. We are now preparing a manuscript to explain more details of Green cGull.
Round 2
Reviewer 3 Report
The authors have responded to my suggestions; I do not have anything further to add.